# Effects of Non-Thermal Treatment on Gilaburu Vinegar (*Viburnum opulus* L.): Polyphenols, Amino Acid, Antimicrobial, and Anticancer Properties

**DOI:** 10.3390/biology11060926

**Published:** 2022-06-17

**Authors:** Berna Erdal, Seydi Yıkmış, Nazan Tokatlı Demirok, Esra Bozgeyik, Okan Levent

**Affiliations:** 1Department of Medical Microbiology, Tekirdag Namik Kemal University, Tekirdag 59830, Turkey; berdal@nku.edu.tr; 2Department of Food Technology, Tekirdag Namık Kemal University, Tekirdag 59830, Turkey; 3Department of Nutrition and Dietetics, Tekirdağ Namik Kemal University, Tekirdag 59030, Turkey; ntokatli@nku.edu.tr; 4Vocational School of Health Services, Adiyaman University, Adiyaman 02040, Turkey; ebozgeyik@adiyaman.edu.tr; 5Department of Food Engineering, Faculty of Engineering, Inonu University, Malatya 44280, Turkey; okan.levent@inonu.edu.tr

**Keywords:** gilaburu, vinegar, antimicrobial, polyphenols, amino acid, ultrasound

## Abstract

**Simple Summary:**

In this study, traditionally produced vinegar made from gilaburu (C-GV) and thermally pasteurized gilaburu vinegar (P-GV), and (ultrasound-treated gilaburu vinegar (UT-GV) were evaluated. At the same time, ultrasound treatment enriched 11 phenolic compounds (gallic acid, protocatechuic acid, hydroxybenzoic acid, vanillic acid, *p*-coumaric acid, rutin, ferulic acid, *o*-coumaric acid, neohesperidin, quercetin, trans-cinnamic). Ultrasound showed different effects on free amino acids and volatile profiles. In general, ultrasound showed more positive results than thermal pasteurization. Six important minerals (Ca, Fe, K, Mg, Mn, and Zn) were detected in gilaburu vinegar, and ultrasound treatment increased the Fe content. Gilaburu vinegar, prepared by different methods, had potential antibacterial and anti-cancer activity.

**Abstract:**

Gilaburu (*Viburnum opulus* L.) is an important fruit that has been studied in recent years due to its phytochemicals and health benefits. In this study, traditionally produced vinegar made from gilaburu fruit (C-GV) was evaluated. Vinegar with higher levels of bioactive components optimized by response surface methodology (RSM) was also produced using ultrasound (UT-GV). The maximum optimization result for the bioactive components was achieved at 14 min and 61.2 amplitude. The effectiveness of thermal pasteurization (P-GV) on gilaburu vinegar was evaluated. An increase was detected for every organic acid with ultrasound treatment. In the UT-GV and C-GV samples, arabinose was present, which is useful for stimulating the immune system. Gilaburu vinegar samples contained 29–31 volatile compounds. The smallest amount of volatile compounds was found in P-GV (1280.9 µg/kg), and the largest amounts of volatile compounds were found in C-GV (1566.9 µg/kg) and UT-GV (1244.10 µg/kg). In the UT-GV sample, Fe was increased, but Ca, K, Mg, and Mn were decreased. A total of 15 polyphenols were detected in C-GV, P-GV, and UT-GV samples, and gallic acid was the most common. A total of 17 free amino acids were detected in gilaburu vinegar samples. Ultrasound provided enrichment in total phenolic compounds and total free amino acids. All three vinegar samples had good antimicrobial activity against pathogens. The efficacy of C-GV, P-GV, and UT-GV samples against colon and stomach cancer was determined, but there were no significant differences between them. As a result, ultrasound treatment is notable due to its antimicrobial and anticancer activity, especially for the enrichment of phenolic compounds and amino acids in gilaburu vinegar.

## 1. Introduction

Acetic acid is a fermented product that forms as a result of the oxidation of bacteria that convert ethyl alcohol to acetic acid [1]. Vinegar has been considered a remedy for bad health since ancient times, and there are various types available on the market. It is used in foods, especially as a spice and salad dressing [2]. The main volatile compound that gives vinegar its unique taste and aroma is acetic acid. Other volatile compounds are esters, acids, alcohols, aldehydes, and ketones [3]. Vinegar has antimicrobial properties that make it a good alternative to antibiotics. At the same time, numerous functional properties of vinegar, such as antimicrobial, antioxidant, antidiabetic, antitumor, anti-inflammatory, antihypertensive, anticancer, and immune stimulant effects, were proven in scientific studies [1,3,4,5].

Studies about alternatives to heat treatment technologies that can guarantee food safety and maintain nutritional quality have been increasing in frequency in recent years. Consumers prefer more healthy and nutritious products. These preferences have encouraged the development of new technologies [6,7]. One of the most commonly used non-thermal technologies in recent years is ultrasound technology. Ultrasound technology is an alternative non-thermal technology to thermal processing that can be applied to many food products, fruit juices, and other products, such as vinegar, to increase the efficiency of unit processes generally applied in the food industry, ensuring food safety by helping to inactivate enzymes and microorganisms [4,8,9,10,11]. It was used for various foods, such as verjuice vinegar [5], tomato vinegar [4], Zhenjiangng vinegar [10], lactic acid fermented mulberry juice [12], red grape juice [13], mango juice [1,4], and citrus fruit juice [14] in recent years.

*Viburnum opulus* L. is a round and red fruit that is especially popular in Eastern Europe and Turkey. *V. opulus* L. is known as dogwood, the American dogwood bush, the cherry tree, the European dogwood bush, the wild eagle rose, and the viburnum rose. The common name used in Turkey is “gilaburu.” Gilaburu fruit and its products have gained popularity in many studies due to their nutritional content, taste, wealth of bioactive substances, and health benefits [15,16,17,18,19]. When the literature was searched, no study was found about ultrasound treatment of gilaburu vinegar. In this study, we investigated how ultrasound treatment of gilaburu vinegar affects its bioactive components as a result of response surface methodology (RSM) optimization. At the same time, antimicrobial and anticancer properties, phenolic compounds, minerals, free amino acids, volatile aroma profile, organic acids, and sugar components in gilaburu vinegar treated with thermal pasteurization and ultrasound were compared.

## 2. Materials and Methods

### 2.1. Preparation of Vinegar

Gilaburu (*Viburnum opulus* L.) fruits from Kayseri, Turkey were the raw material for the production of vinegar. Dark red, fully ripe fruits were selected. They were cleaned of foreign matter and then washed with water. The seeds of the fruits were removed; the red fruit pulp was mixed with deionized water (1:1 w:w) in a blender (Waring Blender, Torrington, Connecticut, USA). Gilaburu vinegar was produced by utilizing the traditional method, as previously described [20]. Periodic acidity measurements were performed, and mothers of vinegar formed on the surface of the gilaburu vinegar at the end of fermentation. Gilaburu vinegar samples were stored at −20 ± 1 °C in 100 mL sterile glass jars for use in analysis. The control (C-GV) sample was untreated traditional gilaburu vinegar. Tests were performed in triplicate.

### 2.2. Thermal Pasteurization and Ultrasound Treatments

Bottles were pasteurized at 85 ± 1 °C in a water bath (Wisd-Model WUC-D06H, Daihan, Wonju, Korea) for 2 min, cooled to 20 ± 1 °C, and labeled pasteurized gilaburu vinegar (P-GV). Ultrasound conditions are shown in Table 1. Ultrasound treatment was applied as in the previous study [9]. Briefly, 100 mL of gilaburu vinegar was processed using a 200 W ultrasonic processor (Hielscher Ultrasonics Model UP200St, Berlin, Germany) at a frequency of 26 kHz. A probe diameter of 10 mm was used in ultrasound treatment. Ultrasound treatments were performed at 26 kHz with a 200 W ultrasonic processor (Model UP200St, Hielscher Ultrasonics, Teltow, Germany) at different amplitudes (40%, 55%, 70%, 85% and 100%) and at different times (2, 5, 8, 11 and 14 min) on C-GV samples. During ultrasound treatments, an ice water bath was used to keep temperature changes below 35 °C, and temperature changes were measured with a thermometer. The vinegar sample was named UT-GV (ultrasound-treated gilaburu vinegar) after optimization. Tests were performed three times.

### 2.3. Experimental Design

The RSM was used to understand the effects of ultrasound treatments on the bioactive components in the gilaburu vinegar. RSM was performed to explore the influences of ultrasound treatment on bioactivity, and results were analyzed by using Minitab Statistical Analysis Software (Minitab 18.1.1 version, Minitab Inc., State College, PA, USA). Optimization conditions are shown according to the central composite design (CCD) (Table 1). Dependent variables were determined as ascorbic acid, and total antioxidant potency (DPPH and CUPRAC), total phenolic content (TPC), total flavonoid content, and total anthocyanin content. The following quadratic-polynomial equation formula was used to create equation models:
(1)$$y=\beta_{0\;}+\sum_{i=1}^{3}\beta_{i}X_{i}+\sum_{i=1}^{3}\beta_{ii}X_{i}^{2}+\sum_{\begin{array}{c} i=1\\ i<j \end{array}}^{3}\sum_{j=1}^{3}\beta_{ij}X_{i}X_{j}$$

The symbols are as follows: the dependent variable (*y*); the intercept term (*β*_0_); the first-order (linear) equation coefficient (*β_i_*); the quadratic equation coefficient (*β_ii_*); the two-factor cross-interaction coefficient (*β_ij_)*; and *X_i_* and *X_j_* are the independent variables.

### 2.4. Determination of Bioactive Compounds

Total phenolic contents of vinegar samples were determined according to the Folin–Ciocalteu method [21]. The measurement was calculated using a standard curve for gallic acid and expressed as milligrams of gallic acid equivalents (mg GAE/L). All analyzes were performed in triplicate. Total flavonoid content was measured with a colorimetric method [22]. Total flavonoid content was expressed as mg catechin equivalents (mg CE/L) per liter. Antioxidant activity was assessed using two different methods: the scavenger 2,2-diphenyl-1-picrylhydrazyl (DPPH) radical and cupric ion reducing antioxidant capacity (CUPRAC) following the methodologies previously described by Grajeda-Iglesias et al. (2016) [23] and Apak et al. (2006) [24], respectively. Trolox equivalent antioxidant capacity (TEAC) standard was used. Total monomeric anthocyanin content (TAC) was determined with the pH differential method described by Giusti and Wrolstad [23]. TAC was determined in triplicate for each treatment, sampling day, and replicate; and results are expressed as mg of cyanidin 3-glucoside equivalents per 100 mL of juice. The ascorbic acid concentration was determined using Tillman’s titrimetric method (2,6-dichlorophenolindophenol sodium) [25].

### 2.5. Analysis of Antimicrobial Activity

*Escherichia coli* (ATCC 25922), *Proteus vulgaris* (ATCC 3851), *Pseudomonas aeruginosa* (ATCC 27853), and *Klebsiella pneumoniae* (ATCC 13883) among Gram-negative bacteria; and *Bacillus cereus* (ATCC 11778), *Staphylococcus*
*aureus* (ATCC 25923), *Enterococcus faecali*s (ATCC 29212), and *Micrococcus luteus* (ATCC 10240) among Gram-positive bacteria, were used as standard microorganisms. Antibacterial activity of gilaburu vinegar samples included in the study were determined using the Kirby–Bauer disk diffusion method in accordance with the recommendations of the Clinical and Laboratory Standards Institute (CLSI) [26]. Bacterial strains were inoculated on 5% sheep blood agar (catalogue number: HM-09912; BES-LAB, Ankara, Turkey) and were then activated by incubating at 35 ± 2 °C for 16–18 h. After incubation, bacterial density was adjusted to 0.5 MacFarland in Mueller Hinton Broth (catalogue number: 4017412; Biolife, Milano, Italia) for all microorganisms. Then, Mueller Hinton Agar (catalogue number: 105437; Merck, Darmstad, Germany) was inoculated with the density-adjusted bacterial suspension. Discs (Bioanalyse BLK, CR, Ankara, Turkey) of 6 mm diameter were impregnated with 100 μL of 6.25–100% C-GV, P-GV, and UT-GV, then placed on to the surfaces of the inoculated plates (90 mm) and incubated at a temperature of 35 ± 2 °C for 16–18 h. Gentamicin (Bioanalyse, 10 µg) discs were used as positive control. Inhibition zone diameters (millimeter) were measured at the end of incubation. Each test was repeated three times.

In this study, the in vitro susceptibility test was performed with the microdilution method in accordance with the Clinical and Laboratory Standards Institute (CLSI) M27-A3 guidelines [27]. *Candida albicans* (*C. albicans*) ATCC 10231 and *Candida parapsilosis* (*C. parapsilosis*) ATCC 22019 were used as standard microorganisms. Serial dilutions were performed 10 times for each C-GV, P-GV, and UT-GV sample concentration varying between 0.20% and 100%. The yeast suspensions were prepared as 1.5 × 10^3^ CFU/mL, and 10 µL of the prepared yeast suspensions were transferred to microplate wells containing different C-GV, P-GV, and UT-GV concentrations. Positive and negative controls were included in each test. The microplates were incubated at 35 °C for 24 h, and the presence or inhibition of microbial activity was determined visually. Minimum inhibitory concentration (MIC) and minimum fungicidal concentration (MFC) values were determined for each isolate. To determine the MFC value, 10 µL aliquots from the wells that were equal two- and three-times the MIC values were transferred to SDA plates. Following incubation at 35 °C for 48 h, the plates were examined for colony formation. The concentration without fungal growth was determined as the MFC value. In the experiment, fluconazole (FLC) and voriconazole (VOR) were used as the reference antifungal agents.

### 2.6. Analysis of Phenolic Compounds

Phenolic compounds were analyzed using an Agilent 1260 Infinity chromatograph, equipped with a diode array detector (DAD). The chromatographic procedure was as described by Portu et al. (2016), using a C-18, ACE Generix column (250 × 4.6 mm; 5 µm packing; Agilent) (Advanced Chromatography Technologies Ltd., Aberdeen, Scotland) [28]. Column temperature was fixed at 30 °C, and the flow rate was 0.80 mL/min. Eluents A and B were used for gradient elution. Solution A was water with 0.1% phosphoric acid, and solution B was acetonitrile. The following gradient was used: 17% B (0 min), 15% (7 min), 20% (20 min), 24% (25 min), 30% (28 min), 40% (30 min), 50% (32 min), 70% (36 min), and 17% (40 min). For the analysis of phenolic compound fractions, the injection volume was 10 µL. Phenolic compounds were identified according to the retention times of the available pure compounds and the UV–Vis data obtained from authentic standards. Detection was carried out at 280, 320 and 360 nm. Concentrations are expressed as μg/mL The results for phenolic compounds are the averages of the analyses of three samples (*n* = 3).

### 2.7. Determination of Organic Acid Contents and Sugar Contents

Organic acid and sugar content, and glucose and fructose contents, were analyzed by high-performance liquid chromatography (HPLC) according to the method proposed by Coelho et al. [29] with minor modifications. The analysis was performed using an Agilent HPLC system, model 1260 Infinity LC (Agilent Technologies, Santa Clara, CA, USA). A sample of 500 μL vinegar was filtered through a 0.45 μm disc syringe filter and a volume of 20 μL was injected. The ion exchange column was an Agilent Hi-Plex H (300 × 7.7 mm). The temperature of the column compartment was maintained at 65 °C, and the RID flow cell was kept at 35 °C. The flowrate applied was 0.6 mL/min for a run time of 20 min. The phase was 10.0 mM/L H_2_SO_4_ in ultrapure water. Standard solutions were injected to obtain the retention time for each compound. For the determination of tartaric, pyruvic, and acetic acids, detection was conducted in the DAD at 210 nm. For the maltose, glucose, turanose, sucrose, xylose, and arabinose sugars, detection was carried out by RID. Results are given as a g/L sample for organic acids and sugars.

### 2.8. Analysis of Minerals

Analysis of Ag (silver), Al (aluminum), Co (cobalt), Cu (copper), Ca (calcium), Fe (iron), Mg (magnesium), Na (sodium), Zn (zinc), P (phosphorus), Ni (nickel), and Pb (lead) amounts in gilaburu vinegar samples was performed with a simultaneous inductively coupled plasma—optical emission spectrometer (ICP-OES) device (Thermo Scientific iCap 6000 Dual view, Cambridge, England). K and Cd content analyses were performed with a flame atomic absorption spectrophotometer (AAS) (Thermo Scientific iCE 3000 Series, Cambridge, England). Dissolution was carried out with a microwave digestion system (Berghof Instruments, Speedwave, Berghof, Germany). [30]. Results are given as mg/L sample for each mineral.

### 2.9. Analysis of Amino Acids

Amino acid content was determined by a method described by Bilgin et al. with slight modifications [31]. Amino acid analysis was performed by using an LC system (Agilent Technologies, Waldbronn, Germany). MS/MS analyses were conducted on an Agilent 6460 triple quadruple LC-MS equipped with an electrospray ionization interface. The JASEM quantitative amino acids kit protocol (Sem Laboratuvar Cihazları A. Ş, Istanbul, Turkey) was used for the determination of amino acid compositions. The samples were read in the device after filtering without acidic hydrolysis and dilution. The results are expressed in mg/100 mL.

### 2.10. Anticancer Activity

A549 (non-small cell lung cancer cells, CCL-185), MDA-MB-231 (triple-negative breast cancer cells, HTB-26), and DU-145 (androgen receptor-positive prostate cancer cells, HTB-81) cells were obtained from American Type Culture Collection and were cultured at 37 °C in 5% CO_2_ and 95% air. All cells were grown in Dulbecco’s modified eagle medium (Gibco, Thermo-Fisher Scientific, Waltham, MA, USA) supplemented with 10% fetal bovine serum (Gibco, Thermo-Fisher Scientific, USA) and 1% streptomycin/penicillin solution. MTT [3-(4,5-dimethylthiazol-2-yl)-2,5-diphenyltetrazolium bromide] method was used to determine the cytotoxic effects of gilaburu vinegar samples obtained by different methods on cancer cells. Cells grown under appropriate culture conditions were removed with the help of Trypsin-EDTA (Sigma-Aldrich, St. Louis, MO, USA), and viable cells were counted in the Thoma cell counting chamber. Cells were seeded on 96-well plates at 1 × 10^4^ cells/well and kept in the incubator for 24 h. Then, vinegar samples were administered to the cells at different concentrations (50, 25, 12.5, 6.25, 3.125%) and incubated for 24 h. Finally, supernatants of cells were removed and washed at least twice with 1 × PBS solution. Then, cells were treated with a 1 mg/mL MTT solution and incubated for 40–60 min at 37 °C. Following incubation, the MTT solution was discarded, and formazan particles were dissolved using DMSO. Plates were read at 550 nm with the help of a Varioscan microplate reader (Thermo-Fisher Scientific, ABD). Experiments were repeated three times at least, and cell viability was calculated using the following formula:Cell viability (%)  =  (OD sample/OD control)  ×  100 (2)

In the cell viability assay, optical density (OD) reflects the number of living cells that remained in the culture following treatments. Anticancer activity of the samples was determined with the help of calculating cell viability following sample treatments.

### 2.11. Analysis of Volatile Compounds

Analysis of volatile compounds of the vinegars was performed with the solid-phase microextraction (SPME) method described by Yıkmış et al. using a GC-MS system (Shimadzu Corp., Kyoto, Japan) [9]. The volatile compounds were then identified according to retention index (RI) by using an n-alkane series (C10–C26) under the same conditions mentioned above. WILEY 8 and NIST 05 mass spectral libraries were used to identify peaks.

### 2.12. Statistical Analysis

All assays were performed in triplicate, and results are expressed as mean ± standard deviation (SD). Data were analyzed by performing one-way analysis of variance (ANOVA) (*p* < 0.05). Statistical analysis was conducted using SPSS 22.0 software (SPSS Inc., Chicago, IL, USA) and SigmaPlot 12.0 Statistical Analysis Software (Systat Software, Inc., San Jose, CA, USA).

## 3. Results and Discussion

### 3.1. Optimization of Bioactive Compounds

Ultrasound is an alternative non-thermal technology used for the enrichment of bioactive compounds in foods and food safety [5,32]. Experimental and predictive results of the bioactive component values in gilaburu vinegar samples at different levels of amplitude and time are given in Table 2. The experimental data obtained were subjected to the second-order polynomial regression model. The results of the RSM optimization via the second-order polynomial regression model for the TPC, TFC, TAC, AA, DPPH, and CUPRAC responses are given in Equations (3)–(8).
(3)$$TPC\;(mg\;GAE/100\;mL)=68.481+4.1739A+0.32248B+0.00706A^{2}+0.000346B^{2}-0.0060399AB$$
(4)$$TFC\;(mg\;CE/L)=8.5177+0.52541A+0.041297B+0.000703A^{2}+0.000037B^{2}-0.007563AB$$
(5)$$TAC\;(mg\;C3G/L)=8.4123+0.12750A-0.010323B+0.001297A^{2}+0.000173B^{2}-0.001904AB$$
(6)$$AA\;(mg/100\;mL)=2.9565-0.18424A-0.018548B+0.004855A^{2}+0.000062B^{2}+0.001443AB$$
(7)$$DPPH\;\left(\%\;Inhibition\right)=32.543+2.276A+0.3854B-0.04921A^{2}-0.001853B^{2}-0.02001AB$$
(8)$$CUPRAC\;\left(\%\;Inhibition\right)=37.51+2.444A+0.3933B-0.04813A^{2}-0.001792B^{2}-0.02252AB$$

Table 3 shows the analysis of variance (ANOVA) for TPC, TFC, TAC, AA, DPPH, and CUPRAC. Linear effects of A (*p* < 0.001) and B (*p* < 0.001) applied to gilaburu vinegar samples on TPC, TFC, TAC, AA, DPPH, and CUPRAC values were found to be statistically significant. Cross interactions of factor A and B in gilaburu vinegars were significant for TPC, TFC, TAC, AA, DPPH, and CUPRAC (*p* < 0.001). Two-way interactions were found to be statistically significant (*p* < 0.001). The R^2^ values of the model used in the study for TPC, TFC, TAC, AA, DPPH, and CUPRAC were found to fit at 99.96, 99.99, 99.57, 99.93, 99.09, and 98.48 levels, respectively (Table 3). The interactions of the variables are graphically represented by the three-dimensional (3D) response surfaces and linear regression, as shown in Figure 1A–D. When TPC, TFC, TAC, DPPH, and CUPRAC models were examined, it was found that A and B factors caused a linear increase in bioactive components. The actual values and the estimated values of the model were found to be compatible, as shown in Figure 1A–D. DPPH and CUPRAC results showed high correlation in Figure 1E,F. At the end of RSM, TPC, TFC, TAC, AA, DPPH, and CUPRAC values were determined to be 97.58 mg GAE/100 mL, 12.20 mg CE/L, 8.83 mg C3G/100 mL, 1.66 mg/100 mL, 54.26%, and 60.36% with 14 min and 61.20 amplitudes, respectively (Table 4). At the end of the optimization, although there was a minimal decrease in the amount of AA compared to the C-GV sample alone, ultrasound treatment was found to preserve AA more compared to the P-GV sample. It was reported that the amount of bioactive components increased after ultrasound treatments were applied to watermelon juice, Chokanan mango juices, Kasturi lime, strawberry juice, lactic acid fermented mulberry juice, purple cactus pear, and vinegar samples [20,32,33,34,35,36]. The increase in the amount of bioactive components with the ultrasound process can be attributed to the breaking of cell walls due to the effect of cavitation pressure, thereby releasing the forms bound to the bioactive ingredients [37]. In this context, it was determined that the literature and our study results are compatible, and the effects of cavitation caused increases. However, it was detected that ultrasound treatment caused a minimal decrease in the amount of AA. Similar effects were found by Tiwari et al. (2009) and Santhirasegaram et al. (2013), who reported a reduction in the amount of ascorbic acid during ultrasound treatments [33,38]. They concluded that the cavitation effect during the ultrasound processes was the cause of the reduction in ascorbic acid content. However, the UT-GV sample experienced less reduction in ascorbic acid content than the P-GV sample, with respect to the C-GV sample.

### 3.2. Analysis of Phenolic Compounds and Amino Acids

Phenolic compounds are natural bioactive compounds. In addition to their antioxidant properties, they are natural bioactive molecules that have gained great interest for their use in various industries. They have interesting properties, such as antimicrobial, anti-inflammatory, blood glucose control, lipid metabolism regulation, weight loss, and antiproliferative activities [39,40]. The polyphenol results of vinegar samples are shown in Table 5. The results show that 15 polyphenols were detected in the C-GV, P-GV, and UT-GV samples. Among these polyphenols, gallic acid content (103.57 ± 0.53 μg/mL) was the highest, followed by ascorbic acid (4.61 ± 0.03 μg/mL) and protocatechuic acid (3.25 ± 0.04 μg/mL). However, no significant difference was observed in ferulic acid content between the treated samples and the control, which could be attributed to the low ferulic acid level in gilaburu. UT-GV had the highest total phenolic compound content. As shown in Table 5, a total of 11 phenolic compounds (gallic acid, protocatechuic acid, hydroxybenzoic acid, vanillic acid, *p*-coumaric acid, rutin, ferulic acid, *o*-coumaric acid, neohesperidin, quercetin, trans-cinnamic) increased in quantity with ultrasound treatment (Table 5). Rutin, a flavonoid compound, increased in quantity after treatment with thermal pasteurization and ultrasound. On the contrary, decreases in the amount of rutin were also reported with ultrasound treatments of grape juice [13]. Similar effects were observed in the study applied to plum (*Prunus salicina* L.) juice [41]. Ultrasound treatment of gilaburu vinegar proved to be superior to thermal pasteurization for preserving and enriching its polyphenols. There was no statistically significant difference in gallic acid between the C-GV and UT-GV samples (*p* > 0.05). In our study, it was determined that the amount of gallic acid increased after ultrasound was applied to strawberry juice [36]. Compared to C-GV, a 4.6% decrease was detected in the amount of ascorbic acid in the UT-GV sample (*p* > 0.05). Similar reductions were reported in kiwi juice [42] and mango juice [33] as a result of ultrasound treatments. As a result of ultrasound treatments, sonochemical reactions can occur that increase the oxidative process and lead to the degradation of ascorbic acid [33]. The amount of hydroxybenzoic acid increased by 0.51 μg/mL in the UT-GV sample. In treatments where ultrasound and ultraviolet were applied together, hydroxybenzoic acid residue was reported in mango juice [43]. The cavitation resulting from ultrasound treatment may lead to disruption of cell walls and ultimately the release of bound polyphenolic compounds, resulting in an increase in polyphenols in the samples [44,45]. At the same time, ultrasound treatments can improve the extraction rate and biosynthesis rate of phenolic substances [46]. As a result, ultrasound increased the polyphenols in gilaburu vinegar.

Free amino acid (FAA) results of vinegar samples are shown in Table 5. The results showed increases in UT-GV (36.49 mg/100 mL) and P-GV (35.23 mg/100 mL) total FAA for all treated samples compared to C-GV (34.86 mg/100 mL). Siddeeg et al. reported similar increases in the amount of FAA with non-thermal technology applied to date vinegar compared to the untreated sample, as in our study [47]. It was reported that protein structures can be altered by partial cleavage of intermolecular hydrophobic interactions related to FAA release [48]. After ultrasound treatment, decreases were observed in the amounts of alanine, aspartic acid, glutamic acid, and ornithine compared to the C-GV sample; increases were detected for other amino acids. While increases were detected for leucine, phenylalanine, serine, and valine after ultrasound treatment, decreases were determined after thermal pasteurization. Wheat plantlet juice showed a significant increase at 30 °C for 20 min after ultrasound treatments, whereas significant decreases were detected for other high parameters in the samples [49]. In our study, ultrasound treatment was generally successful, and more effective results were observed in comparison to thermal pasteurization. A study reported that MW-US (microwave-ultrasound) processed bottle gourd (*Lagenaria siceraria*) juice retained more amino acid content compared to conventionally processed juice [50]. Based on the overall results, ultrasound therapy is considered a beneficial practice for increasing the nutritional value of gilaburu vinegar by increasing amino acids. Further studies are needed to elucidate the detailed mechanism of ultrasound treatment-induced increases in FAA concentration in gilaburu vinegar. However, we predict that the microshock waves generated by cavitation during ultrasound treatments facilitate the release of amino acids.

### 3.3. Minerals, Organic Acid, Sugars

The aim was to investigate the changes in organic acids, sugar components, and mineral elements after the processing of gilaburu vinegar with ultrasound and thermal pasteurization. The results for organic acids, sugar components, and mineral element analyses of C-GV, P-GV, and UT-GV samples are shown in Table 6. Organic acids are considered the most important compounds affecting the general acceptability and organoleptic properties of fruit vinegars [10]. Tartaric acid, pyruvic acid, and acetic acid were detected. Acetic acid was dominant in vinegars, and significant differences were detected in the UT-GV samples; the highest value was 97.10 ± 0.50 g/L. Similar increases in the amount of acetic acid were detected in an ultrasound and pulsed electric field (PEF) study of palm vinegar [47]. Increases were detected in all organic acids with ultrasound treatment. Six sugar components were detected in three vinegar samples. The highest sucrose content was in C-GV samples at 3.37 ± 0.12 g/L. No significant change was detected in the amount of maltose or turanose (*p* > 0.05). The ultrasound process resulted in reductions in all sugar components. In the report by Aadil et al. (2015), the opposite effects were seen: they found significant increases in sugar content, glucose, and sucrose amounts in all sonicated grapefruit juice samples compared to controls [51]. A remarkable sugar component in gilaburu vinegar was arabinose. Sugars such as arabinose and rhamnose in gilaburu were shown to stimulate the immune system [52]. Arabinoses were reduced by ultrasound treatment but not detected after thermal pasteurization.

The results for mineral element analyses of C-GV, P-GV, and UT-GV samples are shown in Table 6. Heavy metals were not detected as a result of the analysis. Six minerals (Ca, Fe, K, Mg, Mn, and Zn) were detected in three vinegar samples. The highest levels of K detected in C-GV, P-GV, and UT-GV vinegars were 2.25, 2.08, and 2.06 mg/L, respectively. Decreases were detected in all minerals with thermal pasteurization. After ultrasound treatment, a significant increase in Fe was detected compared to the C-GV sample. A similar increase was detected in Fe in ultrasound–ultraviolet treatment of mango juice [43]. Decreases in Ca, K, Mg, and Mn were detected after ultrasound treatment. A decrease of 0.09 mg/L was detected in the amount of Mg in the UT-GV sample compared to the C-GV sample. In the report by Aadil et al., they found similar effects and significant reductions in Mg when ultrasound treatment was applied to grapefruit juice samples [51]. For Zn, statistically significant differences were not detected in any samples (*p* > 0.05). It was reported that decreases in the amount of Zn were detected after thermo-ultrasound treatments applied to wheat plantlet juice, but no effect was observed in our study [49]. Changes in minerals may be responsible for the destruction of the cell structure and the transition from cells to solution due to cavitation caused by the effect of ultrasound. This is the first study about the effects of non-thermal and thermal pasteurization on the minerals contained in gilaburu vinegar, so further experimental work is required to understand the precise phenomena.

### 3.4. Analysis of Antimicrobial Activity

Overuse of antibacterial drugs has led to an increase in multi-drug-resistant strains. The difficulties experienced in the treatment of diseases caused by these strains have led researchers to seek alternative treatment methods. Some researchers think that the use of plant extracts and other forms of alternative medical treatment will provide alternatives to antibiotics [53]. Gilaburu (*V. opulus* L.), which is widely used in alternative medicine in Turkey, is an important plant in the pharmaceutical industry due to its high levels of phytocompounds, such as anthocyanins, phenolics, triterpenoids, and vitamins. Numerous studies have been reported showing that bioactive *V. opulus* fruit compounds can function as antimicrobial agents [54,55,56,57,58]. However, no study was found investigating the antimicrobial effects of gilaburu vinegar prepared with the traditional method and the methods used in this study.

In this study, the antibacterial activity of C-GV, P-GV, and UT-GV samples, which were prepared with traditional, pasteurized, and ultrasound methods, at concentrations varying between 6.25 and 100%, was determined by disc diffusion method. At the end of the study, it was found that C-GV, P-GV, and UT-GV samples, which had lower inhibition zone diameters compared to gentamicin used as a positive control, showed antibacterial activity depending on the concentration. An antibacterial effect was detected against all strains except *Enterococcus faecalis* for the first three concentrations (25, 50, 100%) of C-GV, P-GV, and UT-GV samples. It was observed that gilaburu vinegar samples prepared with the three different methods had antibacterial effects on *Proteus vulgaris* strains even at a very low concentration (12.5%). These results show that gilaburu vinegar, prepared by different methods, has potential antibacterial activity (Table 7 and Figure 2).

The broth microdilution method was used to examine the antifungal activities of the GV samples tested. FLC (128 µg/mL) and VOR (8 µg/mL) were used as reference antifungal agents. The MIC and MFC values of C-GV, P-GV, and UT-GV against *C. albicans* and *C. parapsilosis* are given in Table 8. The MIC values tested were 6.25–100% for C-GV, P-GV, and UT-GV. Potent fungistatic effects were detected against *Candida albicans* and *Candida parapsilosis* at 6.25%. While *Candida albicans* had MFC values equivalent to MIC, *Candida parapsilosis’* MFC values were found to be 12.5%. This result shows that the fungicidal (12.5%) effects of all three vinegar samples on *Candida parapsilosis* occurred at higher concentrations than the fungistatic (6.25%) effect. On the whole, C-GV, P-GV, and UT-GV samples were found to have potent antifungal effects against *Candida albicans* and *Candida parapsilosis*. The MIC for FLC against *Candida albicans* was 4 µg/mL, and the MIC for VOR against *Candida parapsilosis* was 0.0625 µg/mL (Table 8).

In another study, it was observed that *V. opulus* L. ethanolic extract had high antifungal activity against *Fusarium* spp. isolated from diseased potato tubers [54]. In a similar study, it was found that the ethanolic extract of *V. opulus* L. had better antimicrobial activity than its aqueous extracts [59]. Another study found that *V. opulus* juices potently inhibited the growth of both Gram-negative and Gram-positive bacteria, whereas higher resistance was found in yeasts [55]. In a study comparing the antimicrobial activities of *V. opulus* L. fruit juices and ethanol extracts with the agar well diffusion method, it was determined that fruit juices had stronger antibacterial activity compared to ethanol extracts. The strongest antibacterial activity was found against *Salmonella typhimurium* (23.6 mm), *Salmonella agona* (20.7 mm), and *Listeria monocytogenes* (19.1 mm). In contrast, the growth of the yeast cultures exhibited little or no sensitivity to the fruit juices and ethanol extracts [53]. In a study conducted in our country, it was found that the aqueous extract of gilaburu had antibacterial activity to various degrees on test microorganisms; however, it was determined that it did not have antifungal activity. Among the tested bacteria, *Pseudomonas aeruginosa* ATCC 9027 was found to be the most sensitive [60]. As a result of another study conducted in our country, it was determined that the ethanol, methanol, and ethyl acetate extracts of *V. opulus* L. fruits have very strong antifungal activity (MIC values; 500–1500 µg/mL) against *Candida* strains (*C. albicans* n:23, non-albicans n:22) that cause urinary tract infections [61].

In conclusion, the results of other studies show that gilaburu extracts prepared in different forms are promising in terms of antimicrobial activities. Since there was no study in the literature investigating the antimicrobial activity of C-GV, P-GV, and UT-GV, our results should be supported by similar studies in future.

### 3.5. Anticancer Activity

In our study, the cytotoxic effects of vinegars obtained from gilaburu fruit by traditional, pasteurized, and ultrasound techniques were investigated in A549, DU-145, and MDA-MB-231 cells. Vinegar samples were found to have different effects on different types of cancer (Figure 3). In particular, dose-dependent inhibition of cell proliferation was observed. Effective doses of C-GV, P-GV, and UT-GV against A549 lung cancer cells were determined to be 36%, 42%, and 27%, respectively. The high anti-cancer activity of UT-GV in A549 cells may be due to its richness of phenolic compounds. In addition, the effective doses of C-GV, P-GV, and UT-GV against MDA-MB-231 cells were determined to be 44%, 40%, and 40%, respectively. Gilaburu vinegar was found to be effective in MDA-MB-231 cells, but no significant difference was found between C-GV, P-GV, and UT-GV. In addition, gilaburu vinegar was found to have anticancer activity at high doses against DU-145 cells: effective doses of C-GV, P-GV, and UT-GV were 61%, 72%, and 63%, respectively.

Previous studies have focused on the juice or extracts of gilaburu fruit, but no study has proved an anti-cancer effect of gilaburu vinegar. The use of natural products isolated from plants in cancer treatment has increased in recent years. Polyphenolic compounds and phytochemicals obtained from plants and their fruits have attracted the attention of researchers, due to their minor side effects, for cancer prevention and alternative treatment. Zakłos-Szyda et. al. (2019) reported that the phenolic-rich fraction obtained from *V. opulus* juice showed potent activity in terms of reducing glucose uptake and free fatty acids in colon cancer cells [62]. *V. opulus* juice has also been shown to have cytotoxic activities against human cervical cancer cells (HeLa) and colon adenocarcinoma cells (Caco-2) [63]. In the same study, experiments were performed with HUVEC endothelial cells to investigate the genotoxic and antiangiogenic effects of *V. opulus* juice, but no observable effect was obtained [63]. Moreover, methanol and acetone extracts of *V. opulus* have been shown to suppress cell migration in MCF-7 and HeLa cells and reduce mitochondrial potential [64]. In addition to in vitro studies, it has been shown that *V. opulus* juice may be beneficial in the prevention of colon cancer induced by 1,2-dimethylhydrazine in Balb-c mice [65]. In addition, *V. opulus* extract has been reported to exhibit anticancer activity in the Ehrlich ascites tumor model [66]. Collectively, these results show that gilaburu vinegar treated by ultrasound has higher anti-cancer activity due to its high content of phenolic compounds. However, further comprehensive studies are needed to demonstrate the in vivo anti-cancer activity of gilaburu vinegar.

### 3.6. Volatile Compounds

Table 9 shows the volatile compounds identified in the C-GV, P-GV, and UT-GV samples. Principal component analysis (PCA) was used to evaluate the differences in volatile compounds of C-GV, P-GV, and UT-GV samples, and clustering analysis was performed. The PCA plot in Figure 4A shows the distribution of samples on two principal components. Eigenvector values in the score graph where all samples are evaluated are 100% (PC1 = 88.7% and PC2 = 11.3%). PCA is suitable for distinguishing gilaburu vinegar samples and grouping volatile compounds according to their spatial locations. C-GV was positively charged on PC1 and PC2; P-GV was negatively charged on PC1 and positively charged on PC2. However, UT-GV was positively loaded on PC1 and negatively loaded on PC2, and was grouped with eight volatile compounds. The dendrogram clusters for C-GV, P-GV, and UT-GV samples are shown in Figure 4B. When the dendrogram was examined, the volatile aroma profiles of the most similar gilaburu vinegars were grouped first, and the starting groups were combined according to their similarities. In cluster analysis, classes according to distances are separated by colors. The red (3), green (3), blue (23), and orange (2) cluster groups are separated.

Gilaburu vinegar samples contained 29–31 volatile compounds, and the most commonly identified groups were alcohols (11), acids (8), and ketones (6). The lowest amounts of volatile compounds were found in P-GV (1280.9 μg/L), and the highest amounts of volatile compounds were found in C-GV (1566.9 μg/L) and UT-GV (1244.10 μg/L) (Table 9). Thermal pasteurization was more effective for change overall. ρ-Cymene and octanal compounds were not detected in the P-GV sample. A decrease in the octanal compound was detected after ultrasound treatment. In reports of different effects on volatile compounds of thermal pasteurization and ultrasound treatment applied to pomegranate juice, ρ-cymene compounds decreased in both treatments, as in our study [67]. Similar effects were also detected after ultrasound was used to remove bitterness in citrus juice [14]. Ultrasound treatment preserved the total aldehyde amounts compared to thermal pasteurization. Similar results were found for thermosonication treatment applied to grape juice: reductions in aldehyde compounds [68]. 6-Methyl-5-hepten-2-one, which provides a significant contribution to the fresh and green sensory properties of most fruits, was detected at 0.72 and 1.09 μg/L in P-GV and UT-GV samples, respectively (Table 9). The changes occurring in aromatic compounds can be explained by the effect created by the micro-shockwaves generated by cavitation during the ultrasound process.

## 4. Conclusions

In this study, the bioactive components in gilaburu vinegar were enriched with ultrasound treatment (except ascorbic acid). The effects were modeled with high RSM optimization. The important parameters of ultrasound-treated giburu vinegar, thermal pasteurization-treated gilaburu vinegar, and untreated gilaburu vinegar were compared. Ultrasound increased the quantities of total phenolic compounds and free amino acids in gilaburu vinegar. Antimicrobial activities have been proven against important bacteria and fungi. Efficacy was observed against MDA-MB-231 and DU-145 cancer cell lines. It was observed that ultrasound treatment affected the mineral and aroma profiles, but was superior to thermal pasteurization. It is concluded that the results of this study should lead to future in vivo studies.

## Figures and Tables

**Figure 1 biology-11-00926-f001:**
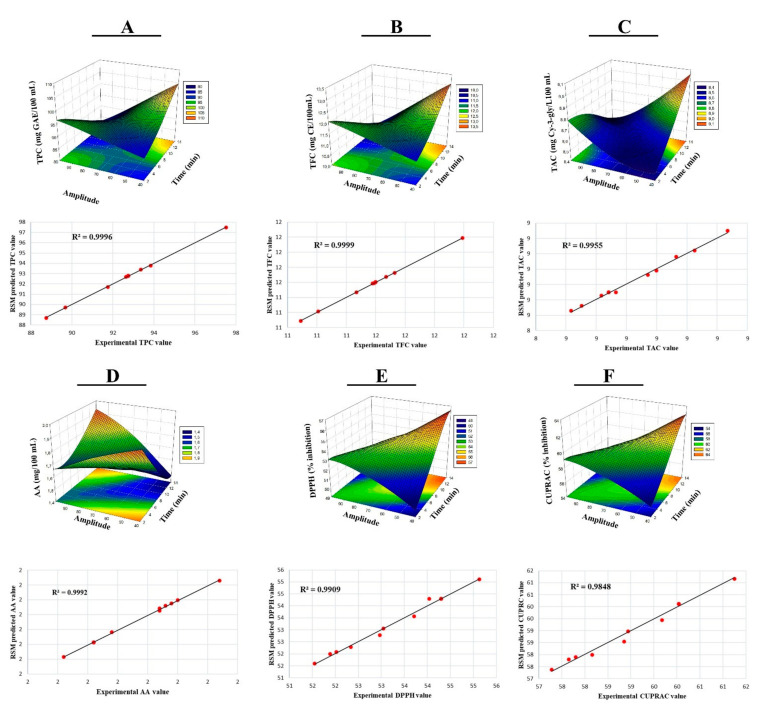
Response surface plots (3D) and linear regression of TPC (**A**), TFC (**B**), TAC (**C**), AA (**D**), DPPH (**E**), and CUPRAC (**F**), as functions of significant interaction factors.

**Figure 2 biology-11-00926-f002:**
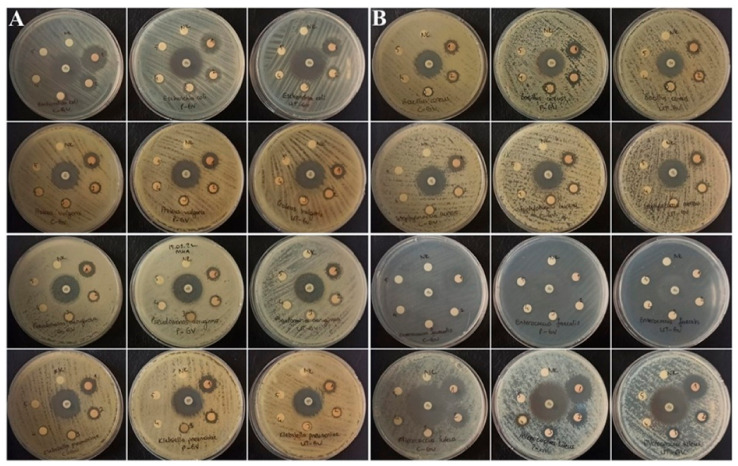
The antibacterial activity of five different concentrations (6.25–100%) of traditional gilaburu vinegar (C- GV), pasteurized gilaburu vinegar (P-GV), and ultrasound-treated gilaburu vinegar (UT-GV) against Gram-negative (*Escherichia coli, Proteus vulgaris*, *Pseudomonas aeruginosa*, and *Klebsiella pneumoniae*) (**A**) and Gram-positive (*Bacillus cereus*, *Staphylococcus aureus*, *Enterococcus faecalis*, and *Micrococcus luteus*) (**B**) bacteria.

**Figure 3 biology-11-00926-f003:**
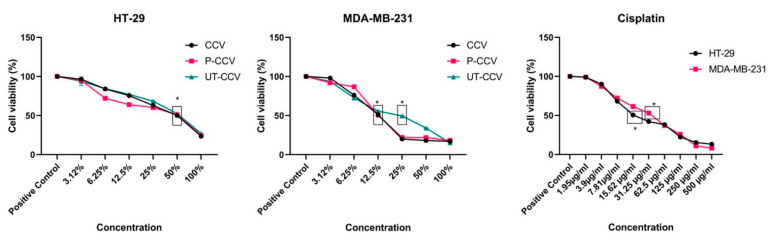
Cytotoxic effects of gilaburu vinegar prepared by traditional, pasteurized, and ultrasonic methods on A549, MDA-MB-231, and DU-145 cells. The vinegars were found to have dose-dependent anticancer effects on different types of cancer cells. * indicates *p* < 0.05.

**Figure 4 biology-11-00926-f004:**
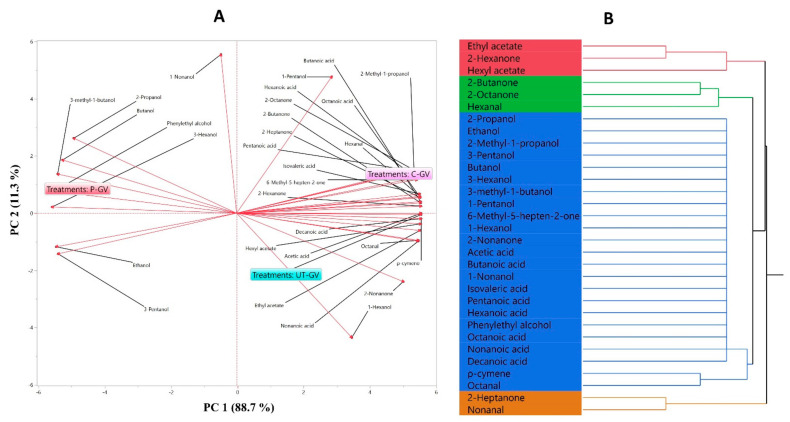
(**A**) PCA bi-plot of volatile compounds in gilaburu vinegar samples. (**B**) Dendrogram of hierarchical cluster analysis of samples and identified volatile compounds. The samples were clustered according to VOCs in the form of red, green, blue, and orange.

**Table 1 biology-11-00926-t001:** Independent variables and their levels in the RSM.

Independent Variable	Factor Levels				
	Lowest	Low	Center	High	Highest
	(−1.41)	(−1)	0	(+1)	(+1.41)
Time (Factor 1, A) (min)	2	5	8	11	14
Amplitude (Factor 2, B) (%)	40	55	70	85	100

**Table 2 biology-11-00926-t002:** Measured responses used in the experimental design for RSM and results of C-GV and P-GV.

Run No.	Independent Variables		Dependent Variables											
	Time (A)	Amplitude (B)	TPC (mg GAE/100 mL)		TFC (mg CE/100 mL)		TAC (mg Cy-3-gly/100 mL)		AA (mg/100 mL)		DPPH (% Inhibition)		CUPRAC (% Inhibition)	
			Experimental Data	RSM Predicted	Experimental Data	RSM Predicted	Experimental Data	RSM Predicted	Experimental Data	RSM Predicted	Experimental Data	RSM Predicted	Experimental Data	RSM Predicted
1	11 (+1)	55(−1)	97.52	97.49	12.19	12.19	8.77	8.78	1.56	1.56	55.13	55.11	61.25	61.16
2	11 (+1)	85 (+1)	88.75	88.69	11.09	11.09	8.56	8.56	1.74	1.74	52.34	52.29	58.16	58.00
3	8(0)	70 (0)	92.75	92.77	11.60	11.60	8.58	8.57	1.61	1.61	54.04	54.29	60.04	60.12
4	2 (−1.41)	70 (0)	92.64	92.67	11.58	11.58	8.53	8.53	1.82	1.81	51.89	52.00	57.65	57.80
5	8(0)	70 (0)	92.75	92.77	11.60	11.60	8.57	8.57	1.61	1.61	54.30	54.29	60.04	60.12
6	8(0)	70 (0)	92.75	92.77	11.60	11.60	8.57	8.57	1.61	1.61	54.30	54.29	60.04	60.12
7	8(0)	70 (0)	92.75	92.77	11.60	11.60	8.57	8.57	1.61	1.61	54.30	54.29	60.04	60.12
8	14 (+1.41)	70 (0)	93.36	93.38	11.67	11.67	8.71	8.71	1.75	1.75	53.05	53.05	58.94	58.97
9	5 (−1)	85 (+1)	93.84	93.77	11.73	11.73	8.65	8.65	1.64	1.64	53.71	53.56	59.68	59.44
10	8 (0)	100 (+1.41)	89.68	89.71	11.21	11.22	8.68	8.69	1.72	1.71	51.55	51.60	57.28	57.38
11	5 (−1)	55 (−1)	91.76	91.70	11.47	11.47	8.51	8.51	1.72	1.72	52.97	52.78	58.85	58.54
12	8 (0)	70 (0)	92.75	92.77	11.60	11.60	8.57	8.57	1.61	1.61	54.30	54.29	60.04	60.12
13	6 (0)	40 (−1.41)	92.72	92.74	11.59	11.59	8.64	8.63	1.73	1.73	52.02	52.08	57.80	57.90
C-GV			92.26		11.10		8.5		2.35		50.92		56.54	
P-GV			90.74		9.68		7.9		1.56		48.60		52.85	

TPC: Total phenolic content; TFC: total flavonoid content; TAC: total anthocyanin content; AA: ascorbic acid; DDPH: radical scavenging activity; CUPRAC: cupric reducing antioxidant capacity; GAE: gallic acid equivalent; CE: catechin equivalent; Cy-3-gly: cyanidin 3-*O*–glucoside; C-GV: gilaburu vinegar; P-GV: thermal pasteurized gilaburu vinegar.

**Table 3 biology-11-00926-t003:** Corresponding *p*-values of linear, interaction, and quadratic terms of regression coefficients obtained by RSM of responses for TPC, TFC, TAC, AA, DPPH, and CUPRAC experiments.

Source	DF	F-Value	*p*-Value	F-Value	*p*-Value	F-Value	*p*-Value	F-Value	*p*-Value	*p*-Value	*p*-Value	F-Value	*p*-Value
		TPC (mg GAE/100 mL)		TFC (mg CE/100 mL)		TAC (mg Cy-3-gly/100 mL)		AA (mg/100 mL)		DPPH (% Inhibition)		CUPRAC (% Inhibition)	
Model	5	3896.75	0.00000	10,322.31	0.00000	320.69	0.00000	2152.32	0.00000	152.89	0.00000	90.57	0.000000
Linear	2	5714.21	0.00000	15115.1	0.00000	314.37	0.00000	710.19	0.00000	80.88	0.00000	54.06	0.000000
A	1	145.71	0.00001	370.8	0.00000	563.6	0.00000	492.07	0.00000	40.64	0.00000	27.78	0.001000
B	1	11,424.47	0.00000	30,222.06	0.00000	105.62	0.00000	1051.49	0.00000	133.6	0.00000	88.75	0.000000
Square	2	34.34	0.00024	62.01	0.00000	358.48	0.00000	3234.75	0.00000	161.59	0.00000	83.31	0.000000
A^2^	1	35.69	0.00056	59.62	0.00000	72.14	0.00000	6420.62	0.00000	219.86	0.00000	114.43	0.000000
B^2^	1	47.74	0.00023	90.56	0.00000	709.4	0.00000	590.74	0.00000	173.04	0.00000	88.05	0.000000
2-Way Interaction	1	12,459.79	0.00000	32,911.97	0.00000	741.54	0.00000	2704.48	0.00000	173.38	0.00000	119.42	0.000000
A*B	1	12,459.79	0.00000	32,911.97	0.00000	741.54	0.00000	2704.48	0.00000	173.38	0.00000	119.42	0.000000
Error	7												
Lack-of-Fit	3	*	*	*	*	2.31	0.21800	*	*	2.19	0.23100	*	*
Pure Error	4												
Total	12												
R^2^		99.96%		99.99%		99.57%		99.93%		99.09%		98.48%	
Adj R^2^		99.94%		99.98%		99.25%		99.89%		98.44%		97.39%	
Pred. R^2^		99.75%		99.90%		96.49%		99.44%		95.66%		89.96%	

A: time; B: amplitude; DF: degrees of freedom; TPC: total phenolic content; TFC: total flavonoid content; TAC: total anthocyanin content; AA: ascorbic acid; DDPH: radical scavenging activity; CUPRAC: Cupric reducing antioxidant capacity; GAE: gallic acid equivalent; CE: catechin equivalent; Cy-3-gly: cyanidin 3-*O*-glucoside. * Doesn’t matter in statistical calculation.

**Table 4 biology-11-00926-t004:** Maximum optimization values according to RSM.

Variable	Setting			
Time (A) (min)	14			
Amplitude (B) (%)	61.2			
Response (UT-GV)	Fit	SE Fit	95% CI	95% PI
TPC (mg GAE/100 mL)	97.58	0.056	(97.4447; 97.7095)	(97.3982; 97.7561)
TFC (mg CE/100 mL)	12.20	0.004	(12.1853; 12.2057)	(12.1817; 12.2093)
TAC (mg Cy-3-gly/100 mL)	8.83	0.007	(8.81751; 8.85173)	(8.81150; 8.85774)
AA (mg/100 mL)	1.66	0.003	(1.65730; 1.67088)	(1.65492; 1.67327)
DPPH (% Inhibition)	54.26	0.157	(53.884; 54.628)	(53.753; 54.758)
CUPRAC (% Inhibition)	60.36	0.213	(59.853; 60.861)	(59.675; 61.038)

TPC: total phenolic content; TFC: total flavonoid content; TAC: total anthocyanin content; AA: ascorbic acid; DDPH: radical scavenging activity; CUPRAC: cupric reducing antioxidant capacity; GAE: gallic acid equivalent; CE: catechin equivalent; Cy-3-gly: cyanidin 3-*O*-glucoside; UT-GV: ultrasound-treated gilaburu vinegar.

**Table 5 biology-11-00926-t005:** Results of phenolic compounds and amino acids of C-GV, P-GV, and UT-GV samples.

Studied Compound		Samples		
		C-GV	P-GV	UT-GV
Phenolic compounds (μg/mL)	Ascorbic acid	4.61 ± 0.03 ^b^	3.66 ± 0.09 ^a^	4.40 ± 0.00 ^b^
	Gallic acid	102.35 ± 0.65 ^b^	93.38 ± 0.09 ^a^	103.57 ± 0.53 ^b^
	Protocatechuic acid	2.08 ± 0.06 ^b^	1.77 ± 0.01 ^a^	3.25 ± 0.04 ^c^
	Hydroxybenzoic acid	0.90 ± 0.42 ^a^	0.85 ± 0.54 ^a^	1.41 ± 0.63 ^a^
	Vanillic acid	1.78 ± 1.62 ^a^	1.55 ± 1.59 ^a^	2.24 ± 2.31 ^a^
	Gentisic acid	0.47 ± 0.66 ^a^	0.35 ± 0.49 ^a^	0.05 ± 0.07 ^a^
	*p*-coumaric acid	0.10 ± 0.02 ^a^	0.05 ± 0.01 ^a^	0.12 ± 0.01 ^b^
	Rutin	0.13 ± 0.18 ^a^	0.23 ± 0.01 ^a^	0.48 ± 0.11 ^a^
	Ferulic acid	0.22 ± 0.01 ^a^	0.13 ± 0.11 ^a^	0.29 ± 0.25 ^a^
	*o*-coumaric acid	0.77 ± 0.01 ^a^	0.56 ± 0.13 ^a^	1.27 ± 0.01 ^b^
	Neohesperidin	0.98 ± 0.03 ^b^	0.81 ± 0.06 ^a^	1.06 ± 0.01 ^b^
	Coumarin	0.02 ± 0.01 ^a^	0.00 ± 0.00 ^a^	0.00 ± 0.00 ^a^
	Quercetin	0.14 ± 0.12 ^a^	0.07 ± 0.04 ^a^	0.20 ± 0.21 ^a^
	trans-cinnamic acid	0.21 ± 0.01 ^a^	0.16 ± 0.01 ^b^	0.23 ± 0.01 ^a^
	Flavon	0.02 ± 0.02 ^a^	0.00 ± 0.00	0.00 ± 0.00
Amino acid content (mg/100 mL)	Alanine	1.49 ± 0.01 ^b^	1.42 ± 0.00 ^a^	1.44 ± 0.00 ^a^
	Arginine	2.38 ± 0.00 ^b^	2.30 ± 0.00 ^a^	2.42 ± 0.00 ^c^
	Aspartic Acid	1.49 ± 0.00 ^c^	1.41 ± 0.00 ^b^	1.37 ± 0.00 ^a^
	Cystine	n.d	n.d	n.d
	Glutamic Acid	1.05 ± 0.03 ^a^	1.09 ± 0.01 ^a^	1.01 ± 0.05 ^a^
	Glycine	0.62 ± 0.06 ^a^	0.66 ± 0.04 ^a^	0.67 ± 0.05 ^a^
	Histidine	0.99 ± 0.01 ^a^	1.03 ± 0.03 ^a^	1.15 ± 0.00 ^b^
	Isoleucine	0.49 ± 0.00 ^a^	0.50 ± 0.01 ^ab^	0.53 ± 0.01 ^b^
	Leucine	2.03 ± 0.07 ^a^	1.92 ± 0.00 ^a^	2.09 ± 0.04 ^a^
	Lysine	1.87 ± 0.00 ^a^	1.88 ± 0.00 ^b^	1.96 ± 0.00 ^c^
	Methionine	0.28 ± 0.00 ^a^	0.29 ± 0.00 ^a^	0.28 ± 0.01 ^a^
	Ornitine	0.64 ± 0.00 ^c^	0.60 ± 0.00 ^b^	0.57 ± 0.00 ^a^
	Phenylalanine	1.20 ± 0.00 ^a^	1.19 ± 0.04 ^a^	1.27 ± 0.03 ^a^
	Proline	17.33 ± 0.06 ^a^	17.95 ± 0.00 ^b^	18.52 ± 0.03 ^c^
	Serine	0.53 ± 0.00 ^b^	0.46 ± 0.01 ^a^	0.54 ± 0.01 ^b^
	Threonine	0.71 ± 0.03 ^a^	0.78 ± 0.00 ^a^	0.75 ± 0.00 ^a^
	Tyrosine	0.78 ± 0.02 ^a^	0.81 ± 0.06 ^a^	0.89 ± 0.01 ^a^
	Valine	0.98 ± 0.02 ^ab^	0.94 ± 0.01 ^a^	1.03 ± 0.00 ^b^
	Taurine	n.d	n.d	n.d

Results are presented as mean ± standard deviation (*n* = 3). Values with different letters on the same line are significantly different (*p* < 0.05). C-GV: traditional gilaburu vinegar; P-GV: pasteurized gilaburu vinegar; UT-GV: ultrasound-treated gilaburu vinegar; n.d: not detected.

**Table 6 biology-11-00926-t006:** Organic acid, sugar, and mineral element analysis results of C-GV, P-GV, and UT-GV.

Studied Compound		Samples		
		C-GV	P-GV	UT-GV
Organic acid content (g/L)	Tartaric acid	0.94 ± 0.11 ^a^	0.92 ± 0.14 ^a^	1.30 ± 0.33 ^a^
	Pyruvic acid	0.09 ± 0.01 ^a^	0.05 ± 0.07 ^a^	0.14 ± 0.00 ^b^
	Acetic acid	62.68 ± 0.66 ^a^	68.50 ± 0.71 ^a^	97.10 ± 0.50 ^c^
Sugar content (g/L)	Maltose	0.84 ± 0.05 ^a^	0.72 ± 0.16 ^a^	0.60 ± 0.02 ^a^
	Glicose	1.24 ± 0.05 ^b^	0.85 ± 0.01 ^a^	0.86 ± 0.04 ^a^
	Turanose	1.92 ± 0.24 ^a^	1.34 ± 0.06 ^a^	1.35 ± 0.09 ^a^
	Sucrose	3.37 ± 0.12 ^b^	2.44 ± 0.11 ^a^	2.63 ± 0.06 ^a^
	Ksilose	1.28 ± 0.02 ^b^	0.83 ± 0.14 ^a^	0.62 ± 0.01 ^a^
	Arabinose	3.52 ± 0.88 ^b^	0.00 ± 0.00 ^a^	1.67 ± 0.15 ^ab^
Minerals (mg/L)	Ag	n.d	n.d	n.d
	Al	n.d	n.d	n.d
	Ca	0.42 ± 0.01 ^a^	0.38 ± 0.00 ^b^	0.36 ± 0.00 ^c^
	Cd	n.d	n.d	n.d
	Co	n.d	n.d	n.d
	Cr	n.d	n.d	n.d
	Cu	n.d	n.d	n.d
	Fe	0.05 ± 0.01 ^b^	0.04 ± 0.00 ^b^	0.39 ± 0.00 ^a^
	K	2.25 ± 0.01 ^a^	2.08 ± 0.01 ^b^	2.06 ± 0.01 ^b^
	Mg	0.44 ± 0.01 ^a^	0.33 ± 0.01 ^c^	0.35 ± 0.00 ^b^
	Mn	0.77 ± 0.01 ^a^	0.69 ± 0.00 ^b^	0.68 ± 0.00 ^b^
	Na	n.d	n.d	n.d
	Ni	n.d	n.d	n.d
	Pb	n.d	n.d	n.d
	Zn	0.03 ± 0.01 ^a^	0.02 ± 0.00 ^a^	0.03 ± 0.00 ^a^

Results are presented as mean ± standard deviation (*n* = 3). Values with different letters on the same line are significantly different (*p* < 0.05). C-GV: traditional gilaburu vinegar; P-GV: pasteurized gilaburu vinegar; UT-GV: ultrasound-treated gilaburu vinegar; n.d: not detected.

**Table 7 biology-11-00926-t007:** Inhibition zones of C-GV, P-GV, and UT-GV in mm.

Bacteria Strains	Zone Diameter (Mean ± SD, mm)												
	C-GV				P-GV				UT-GV				CN
	100%	50%	25%	12.5%	100%	50%	25%	12.5%	100%	50%	25%	12.5%	10 µg/mL
*Escherichia coli*	22.04 ± 0.22	14.02 ± 0.08	10.25 ± 0.34	7.33 ± 0.42	21.03 ± 0.20	13.06 ± 0.20	10.26 ± 0.36	7.03 ± 0.06	15.05 ± 0.13	11.07 ± 0.09	7.11 ± 0.11	ND	21.38 ± 0.45
*Proteus vulgaris*	13.08 ± 0.30	10.27 ± 0.36	8.03 ± 0.23	7.03 ± 0.08	14.03 ± 0.10	10.20 ± 0.20	9.04 ± 0.12	8.03 ± 0.06	12.02 ± 0.28	10.01 ± 0.20	9.02 ± 0.09	7.08 ± 0.11	22.20 ± 0.20
*Pseudomonas aeruginosa*	18.05 ± 0.14	11.35 ± 0.44	10.08 ± 0.10	n.d	10.22 ± 0.57	9.07 ± 0.27	8.07 ± 0.21	7.05 ± 0.13	15.03 ± 0.08	9.10 ± 0.10	8.02 ± 0.09	7.04 ± 0.12	23.05 ± 0.13
*Klebsiella pneumoniae*	15.04 ± 0.12	10.03 ± 0.15	8.00 ± 0.25	7.23 ± 0.32	15.15 ± 0.22	10.33 ± 0.47	7.03 ± 0.10	n.d	10.08 ± 0.10	8.06 ± 0.13	7.02 ± 0.08	n.d	22.15 ± 0.18
*Bacillus cereus*	12.32 ± 0.47	10.31 ± 0.39	7.03 ± 0.10	n.d	14.17 ± 0.22	12.33 ± 0.35	10.05 ± 0.18	n.d	12.28 ± 0.33	9.00 ± 0.14	7.04 ± 0.10	n.d	23.03 ± 0.15
*Staphylococcus aureus*	14.30 ± 0.45	9.08 ± 0.26	8.02 ± 0.09	n.d	12.17 ± 0.21	9.24 ± 0.53	8.15 ± 0.13	n.d	13.11 ± 0.15	9.28 ± 0.35	8.03 ± 0.10	n.d	24.04 ± 0.20
*Enterococcus faecalis*	12.05 ± 0.21	n.d	n.d	n.d	12.05 ± 0.11	n.d	n.d	n.d	11.08 ± 0.17	n.d	n.d	n.d	18.17 ± 0.25
*Micrococcus luteus*	20.62 ± 0.58	16.04 ± 0.21	9.30 ± 0.35	8.25 ± 0.34	21.21 ± 0.48	15.10 ± 0.21	10.05 ± 0.20	8.03 ± 0.10	20.20 ± 0.48	11.36 ± 0.34	8.02 ± 0.08	n.d	33.09 ± 0.30

C-GV: traditional gilaburu vinegar; P-GV: pasteurized gilaburu vinegar; UT-GV: ultrasound-treated gilaburu vinegar; SD: standard deviation; n.d: no diameter; CN: gentamicin.

**Table 8 biology-11-00926-t008:** Antifungal activity of C-GV, P-GV, and UT-GV against *Candida* spp.

Fungi Strains	C-GV (%)		P-GV (%)		UT-GV (%)		FLC (µg/mL)		VOR (µg/mL)	
	MIC	MFC	MIC	MFC	MIC	MFC	MIC	MFC	MIC	MFC
*Candida albicans*	6.25	6.25	6.25	6.25	6.25	6.25	4	8	n.d	n.d
*Candida parapsilosis*	6.25	12.5	6.25	12.5	6.25	12.5	n.d	n.d	0.0625	0.125

C-GV: traditional gilaburu vinegar; P-GV: pasteurized gilaburu vinegar; UT-GV: ultrasound-treated gilaburu vinegar; n.d: no diameter; MIC: minimum inhibitory concentration; MFC: minimum fungicidal concentration; FLC: fluconazole; VOR: voriconazole.

**Table 9 biology-11-00926-t009:** Determination of volatile profiles of C-GV, P-GV, and UT-GV.

Compound	RI	C-GV (µg/L)	P-GV (µg/L)	UT-GV(µg/L)
Ethyl acetate	884	1.66 ± 0.13 ^a^	0.64 ± 0.12 ^b^	1.35 ± 0.13 ^a^
2-Butanone	904	0.72 ± 0.19 ^a^	0.51 ± 0.04 ^a^	0.62 ± 0.04 ^a^
2-Propanol	924	22.90 ± 1.61 ^a^	25.35 ± 3.01 ^a^	22.64 ± 0.96 ^a^
Ethanol	932	102.04 ± 4.79 ^a^	111.51 ± 2.77 ^a^	107.54 ± 2.52 ^a^
Hexanal	1079	1.03 ± 0.20 ^a^	0.44 ±0.06 ^b^	0.74 ± 0.09 ^ab^
2-Methyl-1-propanol	1092	24.01 ± 1.50 ^a^	21.19 ± 1.87 ^a^	22.20 ± 1.75 ^a^
2-Hexanone	1097	1.71 ± 0.23 ^a^	0.91 ± 0.08 ^b^	1.36 ± 0.13 ^ab^
3-Pentanol	1109	34.79 ± 1.88 ^a^	42.12 ± 1.93 ^a^	39.34 ± 2.52 ^a^
Butanol	1157	45.81 ± 2.93 ^a^	55.16 ± 3.26 ^a^	46.47 ± 2.89 ^a^
2-Heptanone	1182	3.49 ± 0.35 ^a^	2.04 ± 0.12 ^b^	2.79 ± 0.18 ^ab^
3-Hexanol	1189	46.73 ± 1.54 ^b^	59.01 ± 1.63 ^a^	51.17 ± 2.76 ^ab^
3-methyl-1-butanol	1205	114.08 ± 6.12 ^a^	130.51 ± 4.16 ^a^	116.82 ± 4.75 ^a^
1-Pentanol	1266	9.42 ± 0.93 ^a^	7.17 ± 0.96 ^ab^	5.79 ± 0.47 ^b^
ρ-Cymene	1275	0.21 ± 0.05 ^a^	n.d	0.13 ± 0.04 ^a^
Hexyl acetate	1281	2.37 ± 0.16 ^a^	0.89 ± 0.13 ^b^	1.87 ± 0.11 ^a^
Octanal	1292	0.54 ± 0.10 ^a^	n.d	0.41 ± 0.13 ^a^
2-Octanone	1299	0.82 ± 0.11 ^a^	0.41 ± 0.07 ^b^	0.56 ± 0.09 ^ab^
6-Methyl-5-hepten-2-one	1342	1.61 ± 0.17 ^a^	0.72 ± 0.19 ^b^	1.09 ± 0.18 ^ab^
1-Hexanol	1353	66.45 ± 3.82 ^a^	61.44 ± 2.38 ^a^	70.95 ± 3.97 ^a^
Nonanal	1392	3.54 ± 0.30 ^a^	1.62 ± 0.19 ^b^	2.79 ± 0.18 ^a^
2-Nonanone	1397	1.46 ± 0.18 ^a^	1.04 ± 0.12 ^a^	1.48 ± 0.13 ^a^
Acetic acid	1458	321.51 ± 9.84 ^a^	237.03 ± 10.58 ^b^	288.09 ± 12.28 ^a^
Butanoic acid	1630	101.75 ± 4.1 ^a^	63.62 ± 5.44 ^b^	84.25 ± 6.10 ^ab^
1-Nonanol	1661	3.93 ± 0.27 ^a^	3.91 ± 0.51 ^a^	3.42 ± 0.66 ^a^
Isovaleric acid	1682	421.05 ± 10.46 ^a^	276.71 ± 26.64 ^b^	348.67 ± 19.49 ^ab^
Pentanoic acid	1735	7.51 ±1.30 ^a^	3.10 ± 0.77 ^a^	4.73 ± 1.05 ^a^
Hexanoic acid	1849	21.11 ± 3.74 ^a^	13.70 ± 1.22 ^a^	16.73 ± 1.82 ^a^
Phenylethyl alcohol	1918	69.37 ± 6.48 ^a^	78.86 ± 5.53 ^a^	71.95 ± 1.94 ^a^
Octanoic acid	2061	30.92 ± 3.34 ^a^	18.57 ± 1.54 ^b^	24.97 ± 3.4 ^ab^
Nonanoic acid	2159	66.95 ± 4.79 ^a^	43.49 ± 3.73 ^b^	61.42 ± 5.37 ^ab^
Decanoic acid	2251	37.49 ± 2.86 ^a^	19.18 ± 2.36 ^b^	30.79 ± 1.88 ^a^
Total (µg/L)				
Esters		4.0	1.5	3.2
Alcohols		466.2	513.4	482.9
Aldehydes		5.1	2.1	3.9
Ketones		9.8	5.6	7.9
Acids		354.6	257.7	313.4
Terpenes		0.2	0.0	0.1

RI: retention index; n.d: not determined; UT-GV: ultrasound treated gilaburu vinegar C-GV: traditional gilaburu vinegar; P-GV: thermal pasteurized gilaburu vinegar. Results are presented mean ± standard deviation (*n* = 3). Values with the different letters on the same line are significantly different (*p* < 0.05).

## Data Availability

Not applicable.

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
