# Peer review of "Effects of Non-Thermal Treatment on Gilaburu Vinegar (Viburnum opulus L.): Polyphenols, Amino Acid, Antimicrobial, and Anticancer Properties"

_biology, 2022, doi:10.3390/biology11060926_

Round 1

Reviewer 1 Report

The paper is well done, too rich in results. It was better to separate the analytical data from biological experiments. However, in

Line 397: insert a space between at and the number 3.37

Line 426-429 How can you justify the high concentration of iron in the UT-CN? Your purpose on the destruction of the cell structure by cavitation isn’t right for this result, it’s better to justify this data in another way.  It’s better don’t write this sentence “The destruction of the cell structure by cavitation caused by 426 the ultrasound effect and the migration from cells into the solution may be responsible for 427 the changes in minerals”, but replace with another more significant or supported by literature

Author Response

(Reviewer 1)

Dear Reviewer

First of all, we would like to thank you for your kind suggestions and comments on our article. The publication of the article is very important for our academic progress. We considered your suggestions and made necessary corrections in the article. The corrections are given below.

Point 1. Line 397: insert a space between at and the number 3.37

Answer: Corrected in text

Point 2. Line 426-429 How can you justify the high concentration of iron in the UT-CN? Your purpose on the destruction of the cell structure by cavitation isn’t right for this result, it’s better to justify this data in another way.  It’s better don’t write this sentence “The destruction of the cell structure by cavitation caused by 426 the ultrasound effect and the migration from cells into the solution may be responsible for 427 the changes in minerals”, but replace with another more significant or supported by literatüre

Answer: Thanks for your suggestion fixed

Reviewer 2 Report

The research article titled Effects of non-thermal treatment on gilaburu vinegar (Viburnum opulus L.): polyphenols, amino acid, antimicrobial, and anticancer properties.  is well supported by the scientific evidences. In my opinion it is suitable for publication after incorporating the following changes.

General comments

1. Write scientific name of the plants and p of statistical analysis in italic. Please change throughout the manuscript.

2. Always use complete name at first and then abbreviate.

3. Citations are not according to the journal style. Please change throughout the manuscript.

4. Language needs improvement throughout the manuscript.

5.  Numbering of table needs correction. Table 1 do not exit and table 5 can be seen on page 2 and 4

Abstract

Line 23. What is C-PV? Is it C-GV? A typing mistake?

Material and methods

Line 81. Replace they with fruits

Line 90, please mention the measurements gaps

Line 96, please delete was

Line 124, please write full name instead of TPC at first

Section 2.4. Please mention the standard used in total flavonoids content analysis and antioxidant activity.

Section 2.5. Write the names of microbes always in italic

Section 2.8. Write full name of the minerals at first and then abbreviate

Section 2.10. Write anticancer activity

Line 232. Remove the scientific name of fruit

What is OD? Mention below the equation. Also describe how you calculate the anticancer activity?

Results

Results of antibacterial activity compared with? Standard drug or control?

Line 328-331. According to results catechin, gallic acid and ellagic acid contents were higher as a result of ultrasound treatment but in Table 5 only gallic acid is mentioned. Please check and correct

Line 335-336, Rutin contents decreased in comparison with? From the table 5, an increase can be observed from traditional vinegar. Please carefully check and correct or re-write.

Line 343-344. A 4.6% decrease in ascorbic acid is a notable reduction but statistical analysis showed no difference? Please check

Line 390-391. Write organic acids like/i.e.,/ including/named

Line 442. Please give your results after this line first and then discuss with previous studies.

Line 464-73. Rewrite the first sentence. Write concentration as between 6.25-100% or give all test doses. At the end of paragraph, it is written potential high antibacterial activity but you mentioned against Enterococcus faecalis a high dose was required?

Line 522. Effective doses in percent related to how much inhibition? Please discuss your results in detail and more clearly. Also describe your results first and then discuss with previous published literature.

Author Response

(Reviewer 2)

Dear Reviewer

First of all, we would like to thank you for your kind suggestions and comments on our article. The publication of the article is very important for our academic progress. We considered your suggestions and made necessary corrections in the article. The corrections are given below.

General comments

Point 1. Write scientific name of the plants and p of statistical analysis in italic. Please change throughout the manuscript.

Answer: Scientific names revised and p-values italicized

Point 2. Always use complete name at first and then abbreviate.

Answer: Corrections made in the text

Point 3. Citations are not according to the journal style. Please change throughout the manuscript.

Answer: corrected citations in the manuscript

Point 4. Language needs improvement throughout the manuscript.

Answer: manuscript language revised and certificate added

Point 5. Numbering of table needs correction. Table 1 do not exit and table 5 can be seen on page 2 and 4

Answer: Table 1 has been added. Corrections were made in the manuscript.

Abstract

Point 1. Line 23. What is C-PV? Is it C-GV? A typing mistake?

Answer: Corrections were made in the manuscript.

Material and methods

Point 1. Line 81. Replace they with fruits

Answer: Corrections were made in the manuscript.

Point 2. Line 90, please mention the measurements gaps

Answer: Production detail has been abbreviated with reference to the previous work.

Point 3. Line 96, please delete was

Answer: deleted

Point 4. Line 124, please write full name instead of TPC at first

Answer: Corrections were made in the manuscript.

Point 5. Section 2.4. Please mention the standard used in total flavonoids content analysis and antioxidant activity.

Answer: Standards added.

Point 6. Section 2.5. Write the names of microbes always in italic

Answer: Names of microbes italicized.

Point 7. Section 2.8. Write full name of the minerals at first and then abbreviate

Answer: clear names of minerals written

Point 8. Section 2.10. Write anticancer activity

Answer: As suggested, we used “anticancer activity” instead of anticancer.

Point 9. Line 232. Remove the scientific name of fruit

Answer: As suggested, the scientific name of fruit was removed.

Point 10. What is OD? Mention below the equation. Also describe how you calculate the anticancer activity?

Answer:  Thank you for these important comments. Explanations for OD and calculation of anticancer activity was provided.

Results

Point 1. Results of antibacterial activity compared with? Standard drug or control?

Answer: Gentamicin (Bioanalyse, 10 µg) discs were used as positive control. This explanation is in section 2.5. In addition, a sentence comparing the inhibition zone diameters of gentamicin and vinegar samples has been added to section 3.5.

Point 2. Line 328-331. According to results catechin, gallic acid and ellagic acid contents were higher as a result of ultrasound treatment but in Table 5 only gallic acid is mentioned. Please check and correct

Answer: Thanks for your suggestion. Revised and corrected.

Point 3. Line 335-336, Rutin contents decreased in comparison with? From the table 5, an increase can be observed from traditional vinegar. Please carefully check and correct or re-write.

Answer: Thanks for your suggestion. Revised and corrected.

Point 4. Line 343-344. A 4.6% decrease in ascorbic acid is a notable reduction but statistical analysis showed no difference? Please check

Answer:  Thanks for your suggestion.  It was checked again, but no statistical difference was observed.

Point 5. Line 390-391. Write organic acids like/i.e.,/ including/named

Answer: Corrections were made in the manuscript.

Point 6. Line 442. Please give your results after this line first and then discuss with previous studies.

Answer: The section 3.5 has been revised according to your suggestions.

Point 7. Line 464-73. Rewrite the first sentence. Write concentration as between 6.25-100% or give all test doses. At the end of the paragraph, it is written potential high antibacterial activity but you mentioned against Enterococcus faecalis a high dose was required?

Answer: Thanks for your suggestion. The first sentence has been changed according to your suggestions. Only pure concentrations (100%) of all vinegar samples were found to be effective against Enterococcus faecalis. On the contrary, antibacterial activity was detected at a lower concentration (100-50-25%) in other bacteria. These results show that vinegar samples have antibacterial effect on all bacteria studied depending on the concentration. The decrease in the inhibition zone diameters as the concentration decreases supports this result.

Point 8. Line 522. Effective doses in percent related to how much inhibition? Please discuss your results in detail and more clearly. Also describe your results first and then discuss with previous published literature.

Answer:  As suggested, we regarding was amended and we first described our results and then discussed the current literature.

Reviewer 3 Report

The manuscript investigates how thermal and ultrasound treatment affect the bioactive compounds, physicochemical, antimicrobial and anticancer properties of gilaburu vinegar. Overall, the manuscript is well written and it has potential for being published by Biology but it needs a major revision before it can be published. Below I havr done some comments (C), questions (Q) or suggestions (S) to improve the manuscript.

Comments:

  1. (S)- Introduction: add information and references for vinegar pasteurization. Explain why vinegar pasteurization is necessary.
  2. (C)- Line 80: Rewrite the sentence "Fully ripe fruits were selected which were dark red" as "Dark red fully ripe fruits were selected" 
  3. (Q)- Line 81: Who the seeds were removed?
  4. (C)- Line 85: Please add information about S. cerevisiae (strain, origin etc)
  5. (Q)- Line 85: what was the initial sugar content of gilaburu juice?
  6. (Q)- Line 87: what was the final ethanol content of fermented product?
  7. (S)- Line 88: Please provide more information about acetic acid content of vinegar, biomass content of the inoculum.
  8. (C)- Rewrite ml as mL
  9. (C)- Please provide information about ethanol and acidity analysis (methods, apparatus etc)
  10. (C)- Line 96: Delete "was"
  11. (Q)- Line 96: what was the size-volume of the bottles
  12. (Q)- Line 99: why amplitude was expressed as %?
  13. (C)- Species of microorganisms and plants are written with italics
  14. (C)- Line 166: Please write the "10^3" correctly
  15. (S)- Figure 1: it will be better to provide a better quality picture
  16. (C)- Please renumber the tables. In the manuscript Tables begin from number 2 and not from number 1
  17. (C)- Please add units on Table 2 
  18. (C)- Page numbers are not correct
  19. (Q)- Line 462: What is 100, 50%?
  20. (S)- Line 475: Please give more information in the figure legend.
  21. (C)- Please write the first time Vibunum opulus and then in the manuscript V. opulus
  22. (Q)- Why volatiles on the table are expressed as μg/kg? I believe that μg/L is more appropriate as unit. 

Author Response

(Reviewer 3)

Dear Reviewer

First of all, we would like to thank you for your kind suggestions and comments on our article. The publication of the article is very important for our academic progress. We considered your suggestions and made necessary corrections in the article. The corrections are given below.

Comments:

Point 1. (S)- Introduction: add information and references for vinegar pasteurization. Explain why vinegar pasteurization is necessary.

Answer:  Commercially available vinegars are often thermally pasteurized. Therefore, we wanted to compare in our study.

Point 2. (C)- Line 80: Rewrite the sentence "Fully ripe fruits were selected which were dark red" as "Dark red fully ripe fruits were selected" 

Answer:  Thanks for your suggestion. Revised and corrected.

Point 3. (Q)- Line 81: Who the seeds were removed?

Answer: Thanks for your suggestion. Revised and corrected.

Point 4. (C)- Line 85: Please add information about S. cerevisiae (strain, origin etc)

Answer: The method has been revised with reference to the previous work.

Point 5. (Q)- Line 85: what was the initial sugar content of gilaburu juice?

Answer: Unfortunately, this measurement was not made within the scope of the study. It was produced using the traditional method.

Point 6. (Q)- Line 87: what was the final ethanol content of fermented product?

Answer: This data was not used in this study. The alcohol content of the final product is in the range of 0.02-0.04%.

Point 7. (S)- Line 88: Please provide more information about acetic acid content of vinegar, biomass content of the inoculum.

Answer:  The method has been revised with reference to the previous work. This data was not used in this study. But the inoculum used is 106 CFU/mL in juice.

Point 8. (C)- Rewrite ml as mL

Answer: ml has been rewritten as mL

Point 9. (C)- Please provide information about ethanol and acidity analysis (methods, apparatus etc)

Answer: Unfortunately, this data was not used in this study. But % Titration acidity is around 4%, ethyl alcohol 0.02-0.04

Point 10. (C)- Line 96: Delete "was"

Answer: Corrections were made in the manuscript.

Point 11. (Q)- Line 96: what was the size-volume of the bottles

Answer: 105 cc Glass Jar

Point 12. (Q)- Line 99: why amplitude was expressed as %?

Answer: The device used in the study is adjusted as amplitude. Amplitude was chosen as the parameter.

Point 13. (C)- Species of microorganisms and plants are written with italics

Answer: Species of microorganisms and plants italicized.

Point 14. (C)- Line 166: Please write the "10^3" correctly

Answer: 10correctly spelled.

Point 15. (S)- Figure 1: it will be better to provide a better quality Picture

Answer: jpeg dpi made 1200.

Point 16. (C)- Please renumber the tables. In the manuscript Tables begin from number 2 and not from number 1

Answer: Table 1 was added to the study. The manuscript was revised.

Point 17. (C)- Please add units on Table 2

Answer: The table was revised again.

Point 18. (C)- Page numbers are not correct

Answer: The manuscript was revised.

Point 19. (Q)- Line 462: What is 100, 50%?

Answer: 100, 50%; concentrations of C-GV, P-GV and UT-GV used in the Kirby-Bauer disk diffusion method. Concentrations added to 2.5 section.

Point 20. (S)- Line 475: Please give more information in the figure legend.

Answer: More information provided in the figure legend.

Point 21. (C)- Please write the first time Vibunum opulus and then in the manuscript V. Opulus

Answer: Corrections were made in the manuscript.

Point 1. (Q)- Why volatiles on the table are expressed as μg/kg? I believe that μg/L is more appropriate as unit. 

Answer:  Thanks for your suggestion. Revised and corrected.
